# The Scrotal Excision of Paratesticular Mesothelioma of the Tunica Vaginalis: A Case Report

**Mohammad Hifzi Mohd Hashim** [1] , **Xeng Inn Fam** [1,*], **Hau Chun Khoo** [1] , **Wan Syahira Ellani Wan Ahmad Kammal** [2] **and Haziq Kamal** [3]

1 Urology Unit, Department of Surgery, Universiti Kebangsaan Malaysia Medical Centre, Cheras, Kuala Lumpur 56000, Malaysia

2 Department of Pathology, Universiti Kebangsaan Malaysia Medical Centre, Cheras, Kuala Lumpur 56000, Malaysia

3 MedCentral Consulting, International Youth Centre, Cheras, Kuala Lumpur 56000, Malaysia

\* Correspondence: xenginn@gmail.com

**Abstract:** Mesotheliomas are malignancies which involve mesothelial cells, and are commonly found in the pleura, peritoneum, pericardium, and (rarely) the testis. We present a case of paratesticular mesothelioma that was excised without the testis. An elderly gentleman presented with a painless right scrotal mass, which appeared clinically benign and separable from the underlying testis. An ultrasound showed an extratesticular lesion adhered to the scrotal wall with a complex hydrocele. An excisional biopsy was conducted, and the Jaboulay procedure was performed on the right testis. Pathological examination revealed mesothelioma, showing focal invasion into the underlying stroma. A post-operative computed tomography (CT) scan evaluation manifested no local or distant metastasis. No further surgery was performed, and no chemotherapy or radiotherapy was offered to the patient. Subsequent clinical examinations and radiological scans carried out during each clinic follow-up for two years showed no new lesion or recurrence.

**Keywords:** scrotal excision; paratesticular mesothelioma; ultrasound; immunohistochemical studies; CT scan

## 1. Introduction

Mesotheliomas are malignancies involving mesothelial cells that normally line the body cavities, including the pleura, peritoneum, pericardium, and testis [1]. Asbestos is the principal carcinogen implicated in the pathogenesis of mesothelioma [2]. However, mesotheliomas of the tunica vaginalis account for less than 1% of cases [3]. The first case of testicular mesothelioma reported was in 1957 by Barbara and Rubino as a malignant non-germ-cell tumor.

Hydrocele, an abnormal water level in between the tunica layers of the scrotum, is the most common presentation of testicular mesothelioma, followed by painless testicular or scrotal mass [3]. Physical examination, radiological evaluation, and blood investigations (including tumor markers such as alpha-fetoprotein ($\alpha$FP), beta-human chorionic gonad-otropin ($\beta$-HCG), and lactate dehydrogenase (LDH)) are important to assist in diagnoses [4]. More specifically, an elevation of more than 40 $\mu$g/L in the $\alpha$FP level indicates the teratocarcinoma of the testis, while an elevation of more than 5 IU/L of $\beta$-HCG indicates the malignancy of the testicular carcinoma [5]. In addition, LDH is a non-specific tumor marker which is produced through the glycolytic activity of the tumor and tumor necrosis due to hypoxia [6]. Like any other testicular tumors, high inguinal orchiectomy is the choice of management [7]; however, it was not performed in this case.

## 2. Case Report

A 69-year-old male non-smoker with an underlying primary hypertension presented to our urology clinic with a painless right scrotal mass and hydrocele. The mass had appeared for six months and was slowly progressively increasing in size. Physical examination revealed a hard and non-tender lobulated mass, sizing around 2 cm × 3 cm on the right scrotum, which was mobile and not attached to the underlying testis. The mass was surrounded with a large hydrocele over the right scrotum. Both testes and the scrotal skin overlying it appeared normal. No lymph nodes were palpable over the inguinal or pelvic region.

An ultrasound (Figure 1) showed multiple lobulated heterogeneously hypoechoic soft tissue lesions adhered to the right scrotal wall with a few calcification and marked vascularity specks seen within. A large right hydrocele with internal moving debris was also observed, separating the lesions from the underlying right testis. In correlation with the levels of alpha-fetoprotein (αFP), beta-human chorionic gonadotropin (β-HCG), and lactate dehydrogenase (LDH) that were not elevated, the initial diagnosis of testicular malignancy was less likely.

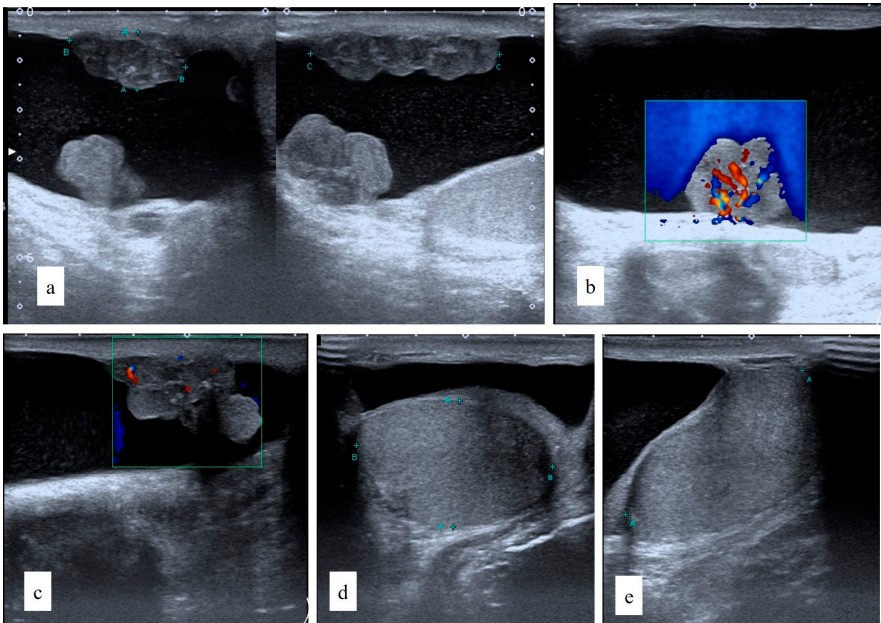

**Figure 1.** Ultrasonography images showing (**a**) right testicular hydrocele with the presence of multiple lobulated heterogeneous hyperechoic lesions adhered to the scrotal wall, (**b**,**c**) marked vascularity seen within the echogenic lesions, and (**d**,**e**) normal right testis free from the lesions.

An excisional biopsy of the scrotal lesion was performed and the Jaboulay procedure was conducted on the right testis. There were multiple lobulated solid masses, gray tan in color, collectively sized at around 3 cm × 3 cm × 4 cm. The scrotal lesions were free from the testis. The hydrocele fluid was clear and straw-colored, and the right testis was normal.

Microscopic examination (Figure 2) showed infiltration by neoplastic mesothelial cells arranged in solid sheets and focal papillary formation with evidence of stromal invasion. The neoplastic cells displayed mild nuclear atypia with vesicular chromatin, conspicuous nucleoli, and rare mitotic activity. There was intermixed infiltration via foamy histiocytes and lymphocytes.

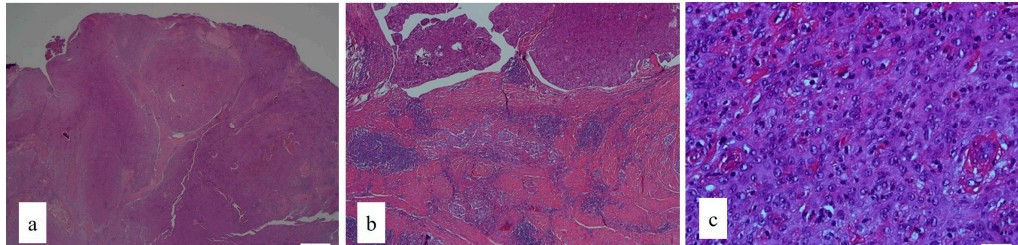

**Figure 2.** Microscopic photograph of the tumor cells showing (**a**) the neoplastic cells arranged in solid sheets (hematoxylin and eosin stain, ×12.5), (**b**) focal papillary architecture and evidence of stromal invasion (hematoxylin and eosin stain, ×40), (**c**) neoplastic cells displaying mild nuclear atypia with vesicular nuclei and conspicuous nucleoli (hematoxylin and eosin stain, ×400).

Immunohistochemical studies (Figure 3) show that neoplastic cells were positive for cytokeratin 7 (CK7), calretinin, and Wilms' tumor gene 1 (WT1), with focal expressions of desmin and epithelial membrane antigen (EMA). They were negative for cytokeratin 20 (CK20), S100 protein, the cluster of differentiation 34 (CD34), the cluster of differentiation 31 (CD31), and smooth muscle actin (SMA). The cluster of differentiation 68 (CD68) highlighted intra-tumoral foamy histiocytes. The overall features were consistent with mesothelioma.

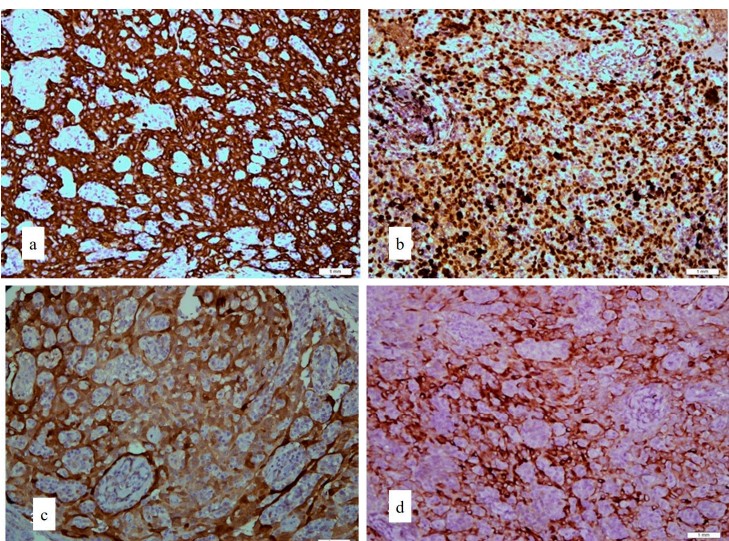

**Figure 3.** Microscopic images showing that the tumor cells were positively stained for (**a**) cytokeratin 7 (CK7) × 200, (**b**) Wilms' tumor gene 1 (WT1) × 200, (**c**) calretinin × 200, and (**d**) epithelial membrane antigen (EMA) × 200.

Computed tomography (CT) of the abdomen, pelvis, and chest was performed two weeks after the operation and showed no sign of regional or distant metastasis and no pathological lymph nodes (Figure 4). The patient refused a right orchiectomy. We consulted with oncologists who suggested no additional treatment. The patient had been on our urology outpatient clinic follow-up for two years, and no disease recurrence was detected clinically and radiologically until the time of this report.

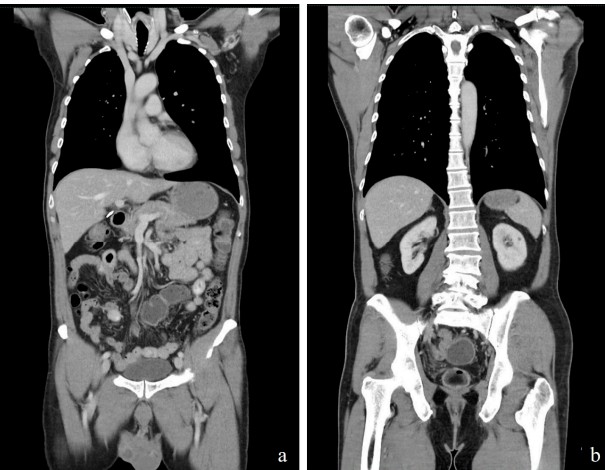

**Figure 4.** Computed tomography scan in (**a**) anterior and (**b**) posterior coronal cuts, showing resolved right hydrocele with no regional pathological lymph node or distant metastasis present.

## 3. Discussion

Mesothelial tumors can arise from any tissue with a mesothelial membrane. Testicular mesothelioma, a testis-specific tumor, develops from the mesothelium covering the tunica vaginalis, tunica albuginea, epididymis, or spermatic cord [2]. Para-testicular mesothelioma is considered an aggressive form of cancer with a mortality rate of 53% over a mean follow-up time of two years [8].

Para-testicular mesotheliomas are distributed over a wide range of ages, but mostly occur in patients in the sixth to eighth decades of life [4]. The developmental mechanism of testicular mesothelioma remains poorly known; however, exposure to asbestos is a well-known risk factor similar any other mesothelioma, and is found in around 35% to 40% of cases of para-testicular mesotheliomas [3]. Trauma, long-term hydrocele, and herniorrhaphy are also attributed as risk factors. The occurrence of testicular mesothelioma is rare and only a few reported cases are found in the medical literature. Thus, the staging system for testicular mesothelioma does not exist due to its rarity [9].

Patients with testicular mesothelioma usually present with scrotal mass and hydrocele as in our case, which can be detected by physical examination. There are many differential diagnoses to exclude, from simple benign conditions to life-threatening testicular malignancies. Blood investigations such as tumor markers including alpha-fetoprotein (αFP), beta-human chorionic gonadotropin (β-HCG), and lactate dehydrogenase (LDH) can aid in the diagnosis of testicular tumors [4]. However, the levels of markers will be normal for cases of mesotheliomas, as seen in our case.

Radiological imaging such as ultrasound, CT, and magnetic resonance imaging (MRI) are essential for the diagnosis, staging, and management of mesothelioma. Ultrasonography is non-invasive and simple, and 90% accurate when used to detect testicular tumors [10]. Positron emission tomography (PET) scans may provide benefits to evaluate for the recurrence of disease [11]. Table 1 shows a summary of clinical features of testicular carcinoma which are described in the case report.

Immunohistochemical staining was performed to aid in accurate diagnosis. Diffuse immunoreactivity for mesothelial markers, including calretinin, cytokeratin 7 (CK7), and Wilms' tumor gene 1 (WT1) was evident [12]. On the other hand, immunostaining for placental alkaline phosphatase (ALP) (a marker of seminomas) and alpha-inhibin (a marker of sex cord stromal tumors) is usually negative [4].

**Table 1.** Summary of the clinical features of testicular mesothelioma cases.

| Case Reports | Clinical Features | | | |
| --- | --- | --- | --- | --- |
| | Age of Male | Physical Examination | Ultrasound Examination | Blood Investigations |
| Gurdal & Erol [8] | 67 | Positive transillumination, right scrotal swelling, and normal left testis and scrotum were detected. | Examination confirmed hydrocele without any suspicion of malignancy. | αFP and β-HCG levels were within normal limits. |
| Goel et al. [13] | 65 | A hard palpable mass was detected in the left iliac fossa and a testicular enlargement was noted on the left side. | Left testis was enlarged 3.9 cm × 3 cm × 3.2 cm, showing diffusely heterogenous echo-texture and irregular nodular surface with irregular hypoechoic thickening of the scrotal wall with left-sided hydrocele. | |
| Park et al. [3] | 65 | A palpable mass was detected in the left inguinal area. The mass was hard, globular, smooth, and nontender, and the lower margins were not palpable below the inguinal ligament. A hard spermatic cord was palpated. The scrotum was normal except for a slightly enlarged nontender left testis. | Ultrasound examination of the abdomen and scrotum showed a 3.0 cm × 3.3 cm × 1.8 cm nodal mass in the left inguinal area. The right testis was 3.0 cm × 2.2 cm × 4.9 cm in size, whereas the left testis was enlarged to 3.3 cm × 2.7 cm × 4.9 cm, contained a little hydrocele, and had normal echogenicity and vascularity. | |
| Akin et al. [4] | 49 | A hard and painless mass was evident in the left scrotum and was suggestive of a hydrocele on palpation. No left inguinal hernia was evident. The right testis was normal, as was the scrotal skin. No palpable lymph node was detected in the pelvic or inguinal areas. | Ultrasonography revealed an increase in the size of the left scrotum, with many multiloculated cysts of different sizes. The parenchyma and size of the left testis were normal. | The levels of αFP, β-HCG, and LDH were not elevated. |
| Kazaz et al. [12] | 75 | A fluid-filled palpable mass (approximately 10 cm in size) filled the left hemiscrotum and extended to the inguinal canal. | Nil | Nil |

Para-testicular mesotheliomas are difficult to manage, and no clear guidelines exist for management purposes. A first-line surgical treatment is inguinal radical orchiectomy for non-metastasis cases [14]. This was not performed in our case as it was not diagnosed at first; however, reported cases suggest that surgical options in the inguino-scrotal area for an early tumor stage are likely to result in better prognosis and smaller rate of recurrence.

Malignant mesotheliomas are aggressive neoplasms capable of widespread local involvement, as well as lymphatic and hematogenous metastases [13]. Para-aortic lymph nodes are primarily and frequently involved in metastasis, followed by iliac, obturator, and inguinal nodes in more advanced stages [3]. The need for adjuvant chemotherapy

or radiotherapy has not been well understood. Cisplatin and pemetrexed can be used as chemotherapies for testicular mesothelioma [15]. Radiotherapy has been shown to be potentially more beneficial than chemotherapy and to be more successful in young patients [12].

In this study, intraoperative images were not provided as we did not expect for the lesion to be a significant tumor. Thus, diagnostic tests such as testicular examination and ultrasound are very important as they may help to diagnose para-testicular mesothelioma, especially for patients presenting with a painless symptom, like in this case report.

## 4. Conclusions

Mesothelioma of the tunica vaginalis of testis is extremely rare but should be kept in mind when diagnosing patients with a testicular or scrotal mass. From the case presented, the mesothelioma was successfully removed via conservative-only surgical management without adjuvant chemotherapy or radiotherapy, suggesting potential management for risky patients that cannot tolerate the adverse effects of chemotherapy or radiotherapy. However, evidence of this is actually scarce and more studies are thus required to demonstrate this hypothesis.

**Author Contributions:** Conceptualization, M.H.M.H., X.I.F., H.C.K., W.S.E.W.A.K. and H.K.; investigation, M.H.M.H. and W.S.E.W.A.K.; interpretation, M.H.M.H., X.I.F., H.C.K., W.S.E.W.A.K. and H.K.; writing—original draft preparation, M.H.M.H. and H.K.; writing—review and editing, X.I.F., H.C.K. and W.S.E.W.A.K.; supervision, X.I.F. All authors have read and agreed to the published version of the manuscript.

**Funding:** This research received no external funding.

**Institutional Review Board Statement:** The study was conducted in accordance with the Declaration of Helsinki.

**Informed Consent Statement:** Informed consent was obtained from the subjects involved in the study.

**Data Availability Statement:** Not applicable.

**Conflicts of Interest:** The authors declare no conflict of interest.

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
