# Peer review of "The Scrotal Excision of Paratesticular Mesothelioma of the Tunica Vaginalis: A Case Report"

_2673-4397, doi:10.3390/uro2040031_

Round 1

Reviewer 1 Report

Dear Authors,

Thank you for this manuscript.

The work focuses an occasional only-surgical conservative management of a paratesticular mesothelioma.

The methodology and the presentation are right. The manuscript is also equipped of clear images.

Nevertheless, the manuscript actually presents some flaws. I would like to offer some suggestions to improve this work. I hope you will share them or, if not, please clarify.

Abstract

Line 11: "Here" may be deleted.

Line 19: please specify the timing of follow up. The abstract should attract the readers if resume perfectly the work.

Introduction

Line 33-37: this part presents diagnostic assessment and typical management, but misses bibliography. Please, add the most updated guidelines in references.

The Authors may also improve the last part of introduction, for example offering a brief diagnostic frame of the expected value of aFP, bHCG and LDH, or typical riadiological framing. In that case, this work may obtain more interest to the readers and more citations.

Case Report

- Since the patient underwent excisional biopsy (not incisional), what about surgical margins? Please specify.

- May you offer intraoperative images?

Discussion

- Since this is an oncological case report, it may be present at least an updated reference to demonstrate that since it is rare, an own staging system does not exist. (see also www.mesothelioma.com/mesothelioma/stages/ or something similar).

The treated case has been not discussed. Please provide (see the comment below).

Conclusions

Line 134: "in conclusion", please remove it. It is just ripetitive.

Line 138-139: this conclusion is not appropriate if compared with the case presented.

I believe that an appropriate conclusion - or a principle for discussion - is that a conservative surgical management without adjuvant chemotherapy or radiotherapy might be considered in the near future (for example in fragile patients, to reduce adverse effects of systemic treatments/radiotherapy), however the evidences are actually scarce and more studies are required to demonstrate this hypothesis. Then, after a brief analysis of this case, you can summarize that <<the surgical orchiectomy remains the modality of treatment...>>. Actually, you did not discuss the case presented, so it lose any scientific power for future developments.

I look forward for your valuable improvements.

Best Regards

Reviewer 2 Report

Nice case report. I consider that once revealed the macroscopic aspect of the tumor is, during surgery, the importance of a radical orchiectomy and the possibility of such a procedure should be underlined to the patient due to the aggressivity of mesotheliomas.

Reviewer 3 Report

The case report is very interesting and very informative. I suggest to include a brief table showing the previous reported cases of urogenital mesothelioma focusing on the tumor prsentation. Finally, please include some points at the end of the manuscript with diagnotic and therapeutic recommendations from managing urogenital mesothelioma.

Round 2

Reviewer 1 Report

Dear Authors,

I appreciated your corrections.

Nevertheless, I found difficult to understand line 106-108. It is grammatically uncorrect. Please fix it.

Line 148-150: I do not understand how diagnostic test represent a limitation in this study, as well as the lack of intraoperative images. I suggest to remove the expression "some limitations" since it appears misleading. You may better explain both two concepts.

Lastly, clonclusions might be improved to stress up the main concept of the presented case (note: this is not mandatory, it is just a suggestion). In particular, line 155-156. Consider to use an expression like "only-surgical management" to stress this concept.

Line 158-159: the expressed concept seems to collide with previously stated. Surgical orchiectomy remains not the modality of treatment, but usually represents the first step in a often multimodal approach. So, you can totally delete line 158-159 (if you agree with me), since the aim/scope of the manuscript have been well described above. 

Thank you for your scientific contribute.

Good luck for your next research!

Best Regards
